# The Role of Mitochondria in Human Fertility and Early Embryo Development: What Can We Learn for Clinical Application of Assessing and Improving Mitochondrial DNA?

**DOI:** 10.3390/cells11050797

**Published:** 2022-02-24

**Authors:** Amira Podolak, Izabela Woclawek-Potocka, Krzysztof Lukaszuk

**Affiliations:** 1Invicta Research and Development Center, 81-740 Sopot, Poland; luka@gumed.edu.pl; 2Department of Obstetrics and Gynecological Nursing, Faculty of Health Sciences, Medical University of Gdansk, 80-210 Gdansk, Poland; 3Department of Gamete and Embryo Biology, Institute of Animal Reproduction and Food Research, Polish Academy of Sciences, 10-748 Olsztyn, Poland

**Keywords:** embryo, embryogenesis, oocyte, oogenesis, fertility, mitochondria, mitochondrial DNA (mtDNA), mitochondrial score, mitochondrial replacement therapy (MRT), autologous mitochondrial transfer

## Abstract

Mitochondria are well known as ‘the powerhouses of the cell’. Indeed, their major role is cellular energy production driven by both mitochondrial and nuclear DNA. Such a feature makes these organelles essential for successful fertilisation and proper embryo implantation and development. Generally, mitochondrial DNA is exclusively maternally inherited; oocyte’s mitochondrial DNA level is crucial to provide sufficient ATP content for the developing embryo until the blastocyst stage of development. Additionally, human fertility and early embryogenesis may be affected by either point mutations or deletions in mitochondrial DNA. It was suggested that their accumulation may be associated with ovarian ageing. If so, is mitochondrial dysfunction the cause or consequence of ovarian ageing? Moreover, such an obvious relationship of mitochondria and mitochondrial genome with human fertility and early embryo development gives the field of mitochondrial research a great potential to be of use in clinical application. However, even now, the area of assessing and improving DNA quantity and function in reproductive medicine drives many questions and uncertainties. This review summarises the role of mitochondria and mitochondrial DNA in human reproduction and gives an insight into the utility of their clinical use.

## 1. Introduction

Mitochondria are well known as ‘the powerhouses of the cell’; however, in recent years, our understanding of its biology has vastly increased. Mitochondria have their own genome which is replicated independently of the nuclear genome. Human mitochondrial DNA (mtDNA) is 16.6 kbp in length and encodes 13 peptides which contribute to all complexes required for oxidative phosphorylation (OXPHOS) except complex 2 [1,2,3]. The remaining mitochondrial proteins are encoded by the nuclear genome and are imported into the mitochondria. The major role of mitochondria is to produce the ATP required by cells. This process relies on OXPHOS whose by-product is the generation of reactive oxygen species (ROS). ROS are generated in approximately 90% of cases by OXPHOS [3]. In addition to this, mitochondria can sequester and release Ca^2+^ regulating calcium responses [4,5]. Moreover, they mediate cell proliferation, differentiation, and apoptosis [3,4,5,6].

Human preimplantation development and embryo implantation is an energy-demanding process that involves a range of energetic cellular processes, requiring significant quantities of ATP [1,2]. Therefore, mitochondria must play a crucial role in proper fertilisation and embryogenesis. This review summarises the role of mitochondria and mtDNA in human oocytes and embryos, showing its influence on fertility and early embryo development. Moreover, we make an insight into the clinical usefulness of assessing and improving mtDNA quantity and function.

## 2. Mitochondria and the Cell Cycle

It is thought that mitochondria are derived from the symbiosis of the prokaryotic α-proteobacteria with the ancient archaea species. From their putative ancestors, they maintain some phenotypic features as a double-membrane, a similar proteome, and the ability to produce ATP via a proton gradient created across its inner membrane [7,8]. However, during the evolution process which led them to the current eukaryotic cells, they lost the capability to synthesize most of the proteins encoded by the primitive bacteria. *Bartonella henselae* is an α-proteobacteria with relatively small DNA homologous to mitochondrial DNA which encodes more than 1600 proteins [7,9]. Additionally, 16,6 kbp mtDNA controls the synthesis of 13 proteins of the OXPHOS, while the rest of the bacterial genes were transferred to the nuclear genome [3,8]. Approximately 1500 nuclear proteins contribute to the mitochondrial proteome, including, e.g., transcription factors, mtDNA polymerase, and ribosomal proteins, as well as enzymes required for the citric acid cycle [6].

The proper inheritance of the genetic material between daughter cells during mitosis requires major cell reorganization. To complete a successful division cycle, the cell follows a specific and coordinated series of events. Moreover, to ensure the correct mitosis, the cellular organelles are also needed to reorganize and segregate [7,10]. Therefore, how do mitochondrial factors influence mitosis? A dynamic equilibrium of fusion and fission is an important feature of the mitochondrial network. The fusion is operated by the dynamin-like GTPases mitofusins 1 and 2 (Mfn1 and Mfn2) at the outer membrane, and optic atrophy 1 (Opa1) localized on the inner membrane. If the cell lacks Mfn1 and Mfn2, no outer membrane fusion occurs [7,8,10,11]. Mice models demonstrated that targeted deletion of either Mfn1 or Mfn2 leads to phenotype consistent with female reproductive aging (e.g., apoptotic cell loss resulting in accelerated follicular depletion). The absence of Mfn1 additionally caused the interruption of oocyte growth and ovulation due to a block in folliculogenesis, whereas Mfn2-lacking oocytes revealed shortened telomeres [12,13,14]. On the other hand, cells lacking Opa1 do undergo outer membrane fusion, however cannot progress to inner membrane fusion. The fission process is mediated by the dynamin-related protein 1 (DRP1) recruited from the cytosol to the outer mitochondrial membrane. Its assembly on the mitochondrial surface causes constriction of the mitochondria and eventual division of the organelle into two separate entities [7,8,10,11]. There are four DRP1 receptors at the mitochondrial outer membrane: Fis1 (Anti-Mitochondrial fission 1 protein), Mff (Mitochondrial fission factor), Mid49 (Mitochondrial dynamics protein of 49 kDa), and Mid51 (Mitochondrial dynamics protein of 51 kDa), from which the latter three play the major role in fission. Repeated cycles of mentioned processes result in the intermixing of the mitochondrial population in the cell and determine mitochondrial morphology. While either increased fusion or decreased fission promotes the formation of elongated mitochondrial networks, increased fission and decreased fusion causes mitochondrial fragmentation [8]. During mitosis, Aurora-A phosphorylates the small Ras-like GTPase RALA (Ras-related protein Ral-A), which localizes to mitochondria and triggers the formation of a complex with RALBP1 (RalA-binding protein 1) and CDK1 (Cyclin-dependent kinase 1)/CyclinB, inducing the phosphorylation of DRP1 to stimulate mitochondrial fission. The knockdown of either RALA or RALBP1 leads to the inhibition of mitochondrial division which results in the inability of even distribution of mitochondria between the daughter cells and, in consequence, in cytokinesis defects [10]. It is noteworthy that experiments conducted with mammalian stem-like cells revealed their ability to asymmetrically sort young and old mitochondria. The daughter cells retaining stem-like features were found to receive most of the new mitochondria, thus, asymmetric portioning of aged mitochondria is required for stemness [15].

## 3. Mitochondrial Genetics

More than 40 years have passed since the first draft human mitochondrial DNA sequence was published [16]. During the last decades, a number of basic and clinical studies were conducted to investigate the mitochondrial genome. 

It is generally accepted that in humans, similarly to most mammal species, mitochondria and mtDNA are exclusively maternally inherited. The paternal mitochondria and their DNA enter the oocyte cytoplasm upon fertilisation; however, the selective elimination of the paternal mtDNA from the oocyte may occur due to some tissue-specific mechanisms of their degradation [17,18,19,20,21,22]. It was demonstrated in mice models that liver mitochondria microinjected into pronucleus-stage embryos were not removed efficiently [19]. On the other hand, the ubiquitination of sperm mitochondria may be the reason for their degradation. It was shown that in some mammals (e.g., rhesus monkeys) the paternal mitochondria in fertilised oocytes are modified with ubiquitin and disappear between the 4-cell stage and 8-cell stage of development. In humans, the sperm mitochondria are tagged with ubiquitin ahead of fertilisation [17,23]. PINK1 (PTEN-induced kinase 1) phosphorylates ubiquitin, and ubiquitin ligase Parkin, leading to the recruitment of Parkin to the outer mitochondrial membrane where it polyubiquitinates multiple proteins. Polyubiquitination of proteins on the outer mitochondrial membrane initiates the recruitment of the machinery that causes the organelle to be engulfed into an autophagosome which, in turn, is directed to lysosomes for degradation, thus leading to the removal of the mitochondria [21,22]. According to another hypothesis, paternal mitochondria are just removed due to the dilution effect derived from far higher numbers of mitochondria in oocytes compared to those in spermatozoa or alternatively, are degraded even before reaching the oocytes [22]. Most likely, the degradation of the paternal mitochondria is regulated by multiple mechanisms.

A crucial role in the maintenance of the maternal inheritance of mtDNA plays the replication of the mitochondrial genome. The population of mtDNA that is inherited is present in the metaphase II oocyte just before fertilisation. A thousand-fold increase in mtDNA copy number is observed—from about 200 copies present in the primordial germ cell (PGC) to over 200,000 copies in mature fertilisable oocytes [24]. Therefore, copies present in the metaphase II oocyte may be either clonal expansion of those in PGCs if the genome is identical or, if the mtDNA variants are present, mutant and wild type molecules can be preferentially selected depending on their frequency and distribution across the oocyte’s cytoplasm (Figure 1) [24]. The presence of more than one mtDNA variant is called heteroplasmy (Figure 1). Apart from the oocyte’s mutations, it may also occur due to either some exceptional cases where paternal mtDNA could be replication-competent and passed to the offspring, or the defect of the above-mentioned mechanisms causing the oocyte unable to remove sperm mitochondria after fertilisation [16,17,25,26,27,28]. It is generally accepted that low levels of mutant mtDNA do not negatively influence the offspring as the coexistence of wild type mtDNA usually neutralises the effect of a mutant load [29]. Some studies reported that around 90% of the individuals in the general healthy population carry at least one mtDNA heteroplasmic mutation [30]. At some levels of mutation content, the offspring may present a mildly affected phenotype [31]. Reaching the threshold of disease-causing mutation frequency leads to mitochondrial dysfunction and, thus, severely affected offspring (Figure 1) [30,31]. Clearly, it is not regarded to every variant in mtDNA, as not all of them is disease-causing. For instance, some of them define the mtDNA haplogroups. Currently, global mtDNA phylogenetic tree contains over 4000 haplogroups [32].

The phenotypic threshold is thought to usually be around 60% for deletions and around 90% for point mutations. However, its specific value depends on the type of mutation and also the tissue being evaluated. For instance, MERF (myoclonic epilepsy associated with ragged red fibres) syndrome occurs only when 90% of mutant mtDNA in muscle is reached. Moreover, experiments conducted directly on individual muscle fibres proved the existence of the phenotypic threshold effect also at the single-cell level [33]. On the other hand, the m.8993T>G mutation in *MT-ATP6* gene reaching the 60% load may result in mild symptoms such as headaches or mild pigmentary retinopathy or drive no effect. Gaining 70–90% of mutation content causes phenotype consistent with NARP (neurogenic muscle weakness, ataxia, and retinitis pigmentosa) syndrome, whereas levels above 90% lead to more severe Leigh syndrome. Contrarily, another point mutation in the same gene, m.8993T>C was demonstrated as less severe, as it affects only those with loads greater than 90% [31]. The phenotypic threshold is rather a phenomenon than a specific value to be obtained. 

An important aspect of mtDNA inheritance is the bottleneck theory [16,26,28]. An absence of mtDNA replication between fertilisation and PGC formation implies a reduction from about 200,000 copies of mtDNA in mature oocytes to about 200 in PGCs. In consequence, cell division and differentiation drive the segregation of mtDNA variants among daughter cells, leading to varying levels within tissues and cells of a given tissue. First of all, the absence of mtDNA replication prevents mtDNA variants from increasing in their frequency. Secondly, it enables selection at the cellular level by enhancing the influence of mtDNA deletions on cell proliferation. Additionally, only a few dozen cells among thousands in the embryo give rise to PGCs, which supports the occurrence of a second bottleneck [28]. The reduction in mtDNA copy number remains until blastocyst formation [24]. Nevertheless, the reduction in mtDNA copy number in PGCs is not the one and only factor that can contribute to the observed germline mtDNA bottleneck. Similarly to population genetics in which the variance among descendants is inversely related to the effective population size, the shifts in frequency of heteroplasmy is observed between generations and also between offspring [34]. 

## 4. Mitochondria in Human Gametes and Embryos

### 4.1. Mitochondrial Distribution in Human Oocytes and Embryos

The heterogeneity in the number of mitochondria present in mature cells is well established. For instance, about 22 to 75 of these organelles are packed around the midpiece in mature human spermatozoa, whereas the high numbers of mitochondria are distributed across the cytoplasm of the oocyte [24]. After fertilisation, the oocyte’s mitochondria are segregated asymmetrically. It seems that a newly fertilised oocyte adjusts mitochondrial density to different intracellular regions; however, its functional significance remains unknown [35,36,37]. It is possible that mitochondrial clustering may serve to supply energy directly and rapidly to the nucleus. This assumption was based on the large size of the oocyte and a hypothesis that the time required for ATP to diffuse across the cytoplasm could be too slow to provide energy [35]. Furthermore, mitochondria are reported to distribute disproportionately among formed blastomeres of developing embryos, though the significance of this phenomenon is also unknown [38].

### 4.2. Mitochondria and Energy Production in the Embryo

As mentioned above, mtDNA does not replicate until the blastocyst stage [24,39]. During that time, the embryo metabolism depends mainly on pyruvate and additionally lactate and amino acids [39,40,41,42,43]. The crucial role of pyruvate as the major energy substrate was confirmed in mice using imaging techniques in living oocytes and embryos. Pyruvate was found to be rapidly metabolised by mitochondria, whereas glucose was not [41,42]. The mitochondrial population in the oocyte must be sufficient to be distributed among formed blastomeres to allow ATP production for functioning until the next mitochondrial biogenesis [39]. However, as it was mentioned before, mitochondria content differs between blastomeres, therefore, the level of ATP production must vary also. 

The important role of glucose starts at the blastocyst stage. Though glucose makes a moderate contribution to ATP production, it becomes a crucial substrate at this point of development [39,40,41,42,43]. Due to increases in glucose as a carbohydrate substrate as well as compacting of mitochondria cristae and initiation of replication, aerobic respiration appears to be upregulated at the blastocyst stage of development. It is estimated that 10% of glucose is metabolised through aerobic respiration in the early stages of development, whereas it increases to 85% in the blastocyst. While the elimination of pyruvate drastically decreases the rate of embryo development, such an effect is not observed for the elimination of either glucose or lactate [41]. Even though aerobic respiration may not be considered to be as essential as anaerobic respiration, the latter cannot be solely sufficient. Hence, both of them are required for proper embryo development.

### 4.3. Mitochondrial Activity versus Fertility and Early Embryogenesis

The importance of mitochondrial activity in proper fertilization and embryogenesis is demonstrated in many aspects. As mentioned above, the developing embryo is in high need of adequate ATP supply. Therefore, insufficient ATP content has been linked to fertilisation failure and abnormal embryo development [39,44]. Actual mitochondrial numbers vary in oocytes, though adequate amounts of mitochondria are required to provide the burst of activity essential up to the blastocyst stage. Thus, mitochondrial dysfunction is revealed when it drops below the necessary threshold. Nevertheless, mitochondrial activity should not be interpreted through the prism of copy number solely as it is strictly regulated by nuclear signals, intracellular ion concentrations, and the availability of substrates [44]. It was demonstrated in the in vitro study that embryonic metabolic response can be quite heterogeneous depending on early phenotypes and the composition of the culture media surrounding the embryos. It showed the ability to switch to other substrates if needed, and to modulate molecular machinery, ensuring their survival even in adverse external conditions [45]. On the other hand, dysfunction of mitochondria in oocytes leading to OXPHOS decrease results in abnormal embryo development [44].

Mitochondrial activity also plays a role in aspects of male fertility. The main role of mitochondria is ATP production from spermatogenesis to fertilization. In addition to this, they modulate several processes, such as spermatogonial stem cells differentiation, testicular somatic cell development, testosterone production in the testis, luminal acidification and sperm DNA condensation in the epididymis as well as ROS homeostasis for sperm capacitation and acrosome reaction in the female reproductive tract [22,46]. Therefore the dysfunction of mitochondria in spermatozoa may also be the reason for infertility.

### 4.4. Effects of Ageing and Other Factors on Mitochondrial Insufficiency

From a reproductive point of view, ageing of females is a progressive decline of ovarian function demonstrated in decreasing quantity and quality of oocytes [47]. It is thought that ovarian ageing may affect oocyte competence by targeting cytoplasmic components including mitochondria [48]. Hence, the quality of mitochondria in the oocyte determines the quality of the oocyte and, in consequence, the developing embryo [47]. However, does mitochondrial dysfunction induce ovarian ageing or ovarian ageing induce mitochondrial dysfunction? 

Experiments both on humans and mice have proved the decrease in mtDNA content in oocytes with ovarian ageing [49,50,51,52,53]. Moreover, a 4977-bp deletion was indicated as another example of a quantitative dysfunction of mtDNA associated with ovarian ageing. Increased frequency of mentioned mutation in oocytes was observed in older women, suggesting the accumulation of this deletion with advancing ageing [48,54]. On the other hand, several studies were performed to evaluate the association of point mutations in mtDNA and ovarian ageing, though the topic remains highly debated [48,52,55,56,57]. Indeed, mtDNA may be more susceptible to damage than nuclear DNA (nDNA) due to its proximity with the free radical-respiratory chain, the lack of protective histones lower fidelity of mitochondrial polymerase γ (POLG), and limited repair mechanisms [48,54,58,59]. As the mitochondrial producing genome, similar to its bacterial ancestors, has no introns and only one major non-coding region (NCR), any point mutation or deletion could disrupt cellular respiration [60,61]. Oxidative stress is considered as one of the factors negatively influencing mtDNA. The hypothesised mechanism involves the damage induced by ROS, a by-product of OXPHOS, which can compromise the integrity of the respiratory chain leading to mitochondria-dependent ageing [47,48,58,59]. ROS are known to react with surroundings proteins, lipids, and DNA, leading to mutations and macromolecule damage [48]. Their mutagenicity was found in the modification of DNA bases. It has been shown that the somatic mutations of mtDNA of older subjects is characterised by a strong G>A mutation preference. Additionally, the ROS-induced formation of mtDNA double-strand breaks seems to be involved in the somatic mtDNA deletions generation [59]. Under physiological conditions, in response, the expression levels of antioxidants enzymes and intracellular proteins rapidly increase [48,58]. However, when ROS are overproduced, these compounds cause oxidative stress and cellular damage. Their high concentration in cells leads not only to mitochondrial and nuclear DNA damage but also apoptosis [58]. Furthermore, conducted experiments found the reducing defense against ROS with ovarian ageing due to the observed age-related down-regulation of genes *SOD1*, *SOD2*, and catalase mRNA encoding key antioxidant enzymes [62]. It is also noteworthy that mitochondria are reported to be targets of the main pituitary gonadotropins. Granulosa cells (GCs) are the main site of steroid hormones synthesis under the control of FSH, and mitochondria have a great contribution in this process. On the other hand, FSH regulates mitochondrial activity via stimulation of mitochondrial biogenesis in GCs under hypoxic conditions. In particular, FSH is considered as a mitophagy-reducing factor playing a crucial role in the maintenance of mitochondrial integrity under oxidative stress conditions. Such an effect can be obtained through FSH-dependent inhibition of the PINK1-Parkin pathway. LH is also regarded as a modulator of the mitochondrial steroidogenic activity and dynamics; however, more studies are needed to fully evaluate its mechanisms. Additionally, mitochondria are involved in oestrogens synthesis, and oestrogens regulate mitochondrial bioenergetics, calcium homeostasis, ROS-scavenger, and their dynamics [63]. Considering the above-mentioned relationship between pituitary–ovarian axis hormones and mitochondrial activity, it seems that age-related hormonal disorders may affect mitochondrial activity, and mitochondrial insufficiency may induce ovarian ageing.

Moreover, excessive Ca^2+^ influx can elevate mitochondrial oxidative stress and lead to apoptosis. Before fertilisation, oocytes at metaphase II stage require the sperm-triggered Ca^2+^ oscillations for several processes such as the resumption of meiosis, polyspermy block, male chromatin decondensation, recruitment of maternal mRNAs, and pronuclear formation. The Ca^2+^ homeostasis in postovulatory oocytes depends on the proper mitochondrial activity as mitochondria-associated membranes facilitate the transfer of Ca^2+^ ions from the endoplasmic reticulum (ER). An excess of Ca^2+^ transfer can either disrupt oxidative phosphorylation and redox homeostasis or trigger the mitochondrial permeability transition pore to open and in consequence affect mitochondrial function and induce apoptosis. That indicates the essentiality of mitochondrial calcium homeostasis for the maintenance of mitochondrial metabolic function; hence, its dysregulation can contribute to pathology [47].

Mitochondrial insufficiency was also reported to result from obesity [64,65]. Due to excessive intake of nutrients, mitochondria become overloaded with fatty acids and glucose, resulting in an increase in the production of Acetyl-CoA. This leads to the production of NADH in the Krebs cycle, which promotes the rise of electrons entering the mitochondrial intermembrane space and in consequence overproduction of ROS inducing oxidative stress. Subsequently, oxidative stress activates several transcription factors including the main mediator of the inflammatory response [65].

The production of ATP is also the primary aspect of mitochondrial function for supporting sperm motility. ROS were found to cause a loss of mitochondrial membrane potential (MMP), lipid peroxidation, impaired sperm motility, and sperm DNA integrity. There are several key pathways through which ROS may be generated by the sperm mitochondria, including disruption of the mitochondrial electron transport, formation of adducts with mitochondrial proteins, reduced mitochondrial expression of prohibitin, the opening of the mitochondrial permeability transition pore (PTP), and induction of apoptosis in spermatozoa. Furthermore, somatic alterations in mtDNA may impair OXPHOS and enhance ROS production, thereby hastening the rate of DNA mutation [60,66].

### 4.5. Impact of Mitochondrial Insufficiency on Fertility

The ageing-related oocyte’s mtDNA copy number reduction is the reason for insufficient ATP levels leading to infertility and abnormal embryo development. Poor oocyte quality is not the only effect of diminished mtDNA content. With decreasing numbers of mitochondria, the proportion of mutant/wild type mtDNA in the oocyte may be increased to not allow the proper embryo development. Such an effect may be an explanation for the surprisingly high number of offspring affected with mitochondrial diseases. Moreover, a decline of ATP levels in the oocyte may be associated with the age-related high risk of aneuploidy. The process of completing meiotic division in the oocyte generally lasts from the preovulatory LH peak to its fertilisation. The proper positioning and segregation of chromosomes depend on the assembly and disassembly of microtubules which is one of the most energy-demanding processes within the oocyte. When oocytes age, their ability to produce adequate spindle microtubules decreases, which in consequence leads to increased incidence of aneuploidy [39,44,58].

Interestingly, mutations in the mitochondria-related genes (e.g., *MT-ATP6* and *MT-CO1*) were linked to primary ovarian insufficiency (POI); however, due to limited sample size, further studies are needed to determine whether mitochondrial dysfunction is a common contributing factor to the onset of POI. If the implication of mitochondrial dysfunction in POI was confirmed, it would indicate the possible therapeutic target for the treatment or prevention of such disorder [67].

Additionally, a recently performed study proved that the alteration of mitochondrial biogenesis of cumulus cells (CCs) could account for the impairment of oocyte quality observed in diminished ovarian reserve (DOR), which also suggests its major role in the determination of oocyte competence. If so, mitochondrial characteristics of CCs could serve as indicators of oocyte competence, and thus oocyte quality in DOR patients may be improved with mitochondrial biogenesis-enhancing therapies [68]. 

In the aspect of male fertility—the motility of human spermatozoa is entirely dependent on the functionality of the OXPHOS pathways. As sperm mtDNA partially encodes for OXPHOS-related proteins, any aberration in the mitochondrial genome may negatively influence sperm motility [60,66]. In addition to this, point mutations and deletions in mtDNA were linked to asthenozoospermia and oligoasthenozoospermia [69]. Furthermore, human spermatozoa with low MMP are less capable of undergoing the acrosome reaction and additionally show a direct and significant correlation with decreased sperm count, and negatively affected morphology, motility, and viability [60,66]. Probably, evaluation of the role of mitochondria in spermatogenesis may reveal new causes of male infertility. Moreover, when taking the effect of ageing, obesity, and metabolic health on the mitochondrial insufficiency into consideration, it is clear that infertility related to poor lifestyle may be settled into mitochondrial causes [66].

For both men and women, the pathogenesis of mitochondrial-related infertility may be direct through germ cells damage or indirect via diminished gonadotropin drive [70].

### 4.6. Do mtDNA Mutations Influence Early Embryo Development?

mtDNA mutations are a frequent cause of severe metabolic disorders. However, coexistence of wild type mtDNA usually neutralises the effect of a mutant load allowing normal phenotype to be maintained [29]. During early embryo development mtDNA content remains stable [1]. Therefore, the presence of mtDNA mutation in an oocyte or a preimplantation embryo may result in the reduction in the absolute number of wild type mtDNA copies. The protection against transgenerational transmission may be obtained either via inducing fertilisation defects and/or early embryogenesis failure or mechanisms eliminating damaged mitochondria during early embryo development. 

The preimplantation mouse embryos were found to eliminate damaged mitochondria through a purifying selection during early embryogenesis; however, the precise mechanisms remain unknown. Such a phenomenon also suggested to take place to counter expansion of deleterious mtDNA mutations in the female germline [71]. On the other hand, a recently conducted study demonstrated that the presence of a pathogenic mutation was not the mtDNA metabolism modifier in human cleavage-stage embryos suggesting both no impact of mtDNA mutations and the absence of selection against them at this stage of development [29]. Contrarily, another study, focused on mtDNA mosaicism in early human development, identified a subgroup of low-level variants that may give rise to stable lineages of genetically diverse cells in the adult due to above-mentioned asymmetrical distribution of mitochondria within the oocyte [72].

Currently, the influence of mtDNA abnormalities on proper fertilisation and subsequent embryo development is not well known. Further studies (both animal and clinical) are needed to fully evaluate the relationship of mtDNA sequence and successful human embryogenesis.

## 5. Clinical Usefulness of Assessing and Improving Mitochondrial DNA Function and Quantity

### 5.1. Assessment of mtDNA Content in Human Embryos and Its Clinical Significance

The importance of mtDNA levels in the fertilisable oocyte was already demonstrated above. However, is there any relationship of mtDNA copy number with the embryo’s ability to successfully implant and develop? In the past, dozens of experiments were performed to evaluate the significance of mtDNA content in developing embryos. The parameter, commonly called mitochondrial score, was defined as the ratio of mitochondrial/nuclear DNA copy number. However, almost every study group differently calculated its value [38,73,74,75,76,77,78,79,80,81,82,83,84,85,86,87,88,89,90,91]. Even now, there are a lot of controversies as conducted studies have reached conflicting results (Table 1).

Most study groups assessing the relationship of mtDNA content and embryo’s ploidy status agree that increased levels of mtDNA correlate with aneuploidy [38,73,76,78,87,89]. Nevertheless, the mechanisms leading to such an effect are still not known. It may result from indirect indication of higher energy needs of aneuploid embryos, possibly stemming from the initiated repair mechanisms [38]. 

The primary aim of conducted experiments was to evaluate the link of mtDNA levels and the embryo’s ability to implant [38,73,74,75,77,78,79,80,81,82,83,84,85,86,87,91]. Indeed, the proof of such a relationship would be a revolutionary discovery, giving the possibility to identify viable embryos and in consequence significantly improve the pregnancy rates of in vitro fertilisation treatment. Nevertheless, obtained results are highly conflicting [38,73,74,75,76,77,78,79,80,81,82,83,84,85,86,87,88,89,90,91]. Mitochondrial score was demonstrated to have either positive [85], negative [74,78,80,83,91] or no correlation [38,73,75,77,82,84,86,87] with implantation rate. A few research teams evaluated the link between mtDNA content with live birth ratio; however, most of them found no statistically significant difference between mitochondrial score values among embryos leading to live birth and those that did not [38,73,74,84,86,90]. Only twice were lower mtDNA copy number values demonstrated to correlate with live birth rate [74,90].

Divergent results were also obtained for assessment of the relationship of the mitochondrial score with embryos’ morphology and especially maternal age [38,73,74,75,76,77,78,79,80,81,82,83,84,85,86,87,88,89,90,91].

### 5.2. Mitochondrial Score—The Debate under Its Usefulness as Embryo Selection Marker

Even though the mitochondrial score was evaluated numerous times by several study groups, its clinical usefulness remains unclear [38,73,74,75,76,77,78,79,80,81,82,83,84,85,86,87,88,89,90,91]. For the first time it was proposed as an embryo selection marker by Fragouli et al. [78]. Although they used both Next-Generation sequencing (NGS) and quantitative RT-PCR (qRT-PCR), and the mitochondrial score count was normalised based on the GC content and in-silico reference as well as taking into consideration the numbers of chromosomes, their study seems to be questionable. Interestingly, when assessing both TE and blastomeres they found positive correlation for mtDNA content with maternal age for TE and negative for blastomeres (Table 1). Such findings seem unlikely, and the reliability of these results should be reconsidered. Additionally, their subsequent study was claimed to echo their first findings, yet the observed relationship was statistically insignificant (Table 1). Even though the correlation for this parameter was not obtained again, they did not decide to reanalyse the rest of parameters, but only evaluated the pre-established thresholds [78,79]. The link of lower mtDNA levels with embryos’ implantation potential was subsequently demonstrated by Diez-Juan et al. [83], Ravichandran et al. [80], Du et al. [91], and Wang et al. [74]. Diez-Juan et al. [83] assessed mtDNA content with qRT-PCR based on *ATP8* gene fragment representing mtDNA and β-actin gene fragment representing nuclear DNA. They did not take the ploidy status of chromosome 7 under consideration, which might influence results obtained for embryos with eventual trisomy or monosomy of chromosome 7. The correlation of mitochondrial score with implantation status was demonstrated for both TE and blastomeres (Table 1). Ravichandran et al. [80] conducted a study using NGS and qRT-PCR techniques. In fact, they did not assess the correlation between mtDNA content with implantation potential, but used the threshold established by Fragouli et al. [78], revalidating it due to technical changes in the laboratory. They concluded 100% negative predictive value of mtDNA assessment based on 33 embryos containing elevated levels of mtDNA that did not produce pregnancy. The sample size seems far too small to draw such conclusions. Both Wang et al. [74] and Du et al. [91] used NGS to evaluate mtDNA content at the blastocyst stage of development. Wang et al. [74] applied the formula proposed by Victor et al. [77], taking into account variables such us ploidy status of each chromosomes and embryo’s genetic sex. They found a statistically significant difference of mitochondrial score values depending on implantation rate and live birth rate (Table 1). However, this relationship was only observed for day-5 biopsied embryos, and no correlation was reported for day-6 biopsied blastocysts. Moreover, the mitochondrial score was doubted as an independent embryo selection marker due to receiver operating characteristic (ROC) curve analysis outcomes, performed to evaluate the potential predictive value of mtDNA content in embryo implantation and live birth outcomes. Contrarily, Du et al. [91], who applied the formula proposed by Shang et al. [82], regarded ROC analysis outcomes as a highly predictive value. Nevertheless, a future, multicentre study was suggested to evaluate the mtDNA content in developing embryos and to verify its clinical application. On the other hand, Wu et al. [85] who used MitoCalc analyzer software dedicated to sequencing coverage analysis, found a positive correlation of mitochondrial score with implantation rate. In fact, obtained results may differ depending on biopsy day of assessed embryos. Day-5 biopsied blastocysts are reported to have higher mtDNA content compared to day-6 biopsied embryos [74,75,84,85]. Thus, the number of embryos biopsied at day-5 or 6 may influence obtained results. Apart from those mentioned above, the rest of listed study groups found no statistically significance of mitochondrial score in the context of assessing implantation potential [38,73,75,77,82,84,86,87]. Moreover, even though most study teams agree that mitochondrial score correlates with ploidy status, it seems unlikely to find the use in such assessment as preimplantation genetic testing for aneuploidies (PGT-A) gives much more definitive diagnosis. Furthermore, evaluation of mtDNA content may be useful at the blastocyst stage of development as the cleavage-stage embryos were demonstrated to present asymmetrical distribution of mitochondria among the formed blastomeres. Therefore, as the embryo may present the heterogeneity of their blastomeres, the obtained results may not be reliable [38]. Indeed, during our recent study we observed extreme differences of the mtDNA levels for embryos from the same patient, which suggests that the choice of a blastomere for biopsy strongly affects the outcomes of mtDNA quantification. In consequence, the use of an arbitrary threshold of mtDNA content as embryo selection criteria, at least for cleavage-stage embryos, must be avoided [38].

As so many study teams evaluated the clinical significance of mtDNA content in early embryos during past years, one question arises—how is it possible that we still have no sure answer? First of all, conflicting outcomes may result from different technical approaches on detection of mtDNA content and used formulas to calculate it. Some study teams take ploidy status for consideration, whereas some do not [77,78,82,83]. Technical approaches seem to play a crucial role as even study groups who used the same formula obtained conflicting results (Victor et al. [77] versus Wang et al. [74] or Shang et al. [82] versus Du et al. [91]) (Table 1). For instance, there are several whole genome amplification (WGA) protocols which differently influence the amplification of mtDNA in comparison to nDNA [38]. Secondly, as mentioned above, the day of biopsy may additionally influence obtained results. Moreover, storage protocols or reagents used for culturing may affect the assessment as the incidence of blastocysts with elevated mtDNA was reported to vary widely between clinics in which in vitro fertilisation was undertaken [80]. Last but not least, the sample sizes vary between conducted research, and mostly it is too small when assessing blastomeres biopsied from cleavage-stage embryos; thus, it cannot deliver reliable outcomes [78,87]. 

What should be done then? The best way to investigate the clinical significance of mtDNA content in developing embryos seems to be the assessment of it in a large, multicenter study using modern laboratory techniques and precise calculations.

Although the clinical value of mtDNA content assessment is still not established well, it is commercially available [92], driving another question—does commercialisation overtake science in reproductive medicine?

### 5.3. Opportunities to Improve Mitochondrial DNA Function

The fundamental role of proper mitochondrial activity and functionality for human health and reproductive success was already summarised. Hence, are there any possibilities to improve mitochondrial DNA function? In the gene-editing era, mitochondrial genome editing (MGE) seems to be one of the possible solutions. MGE refers to the modification of human oocytes or embryos with intracytoplasmic microinjection or mitochondria-targeted nucleases in order to prevent transmission of mitochondrial diseases [93]. Yahata et al. [94] engineered platinum TALENs, which were transported into mitochondria, recognized the mtDNA sequence including the m.13513 position, and preferentially cleaved G13513A mutant mtDNA, and conducted an experiment with induced pluripotent stem cells (iPCs). The heteroplasmy level of m.13513G>A mutation was decreased short after the transduction. Moreover, Yang et al. [95] demonstrated successful elimination of m.3243A>G mutation in iPCs via injection of mitoTALEN mRNA. However, protein engineering and assembly processes for every genomic target seem to be a time-consuming task [93]. On the other hand, CRISPR/Cas9 genome editing technology was also suggested to target mutant mtDNA [96]. The issue of mitochondrial genome editing is definitely of increasing interest to scientists, although there is still a long way to go to introduce it in clinical practice. Nevertheless, there is already a way to overcome the transmission of maternally-inherited mtDNA mutations—the use of mitochondrial replacement technologies in assisted reproductive treatment (ART). Currently, there are two transfer techniques in clinical application—maternal spindle transfer and pronuclear transfer (Figure 2).

Maternal spindle transfer refers to removing the nuclear DNA from the donor egg, leaving the part of the oocyte containing healthy mitochondria, and then inserting into this cell nuclear DNA from the mother’s oocyte to finally fertilise it and implant into the mother’s uterus. The pronuclear transfer is a similar process; however, it requires fertilisation of both—mother’s and donor’s—oocytes before transferring the nuclear DNA [97,98,99,100].

### 5.4. Mitochondrial DNA Transfer in Improving the Reproductive Potential of Oocytes

Mitochondrial replacement therapy is also an object of interest in the context of improving the quality, and hence the reproductive potential, of oocytes [100,101,102]. In 1997 it was performed for the first time using partial ooplasm transfer, a technique based on simultaneous injection of sperm with 1 to 5% of oocyte cytoplasm during intracytoplasmic sperm injection (ICSI) [103,104,105]. The ooplasm was derived from fresh or cryopreserved oocytes of young and healthy donors, and clinically abandoned trinuclear embryos. Although this technology achieved good clinical results, it was banned by the US Food and Drug Administration (FDA) for clinical use in the US due to the ethical and genetic safety issues caused by the involvement of a third-party genetic material [102]. A potential alternative approach could be autologous mitochondrial transfer, theoretically introducing much larger amounts of mitochondria into the oocyte. Mitochondria preferentially should be obtained from ovarian or oocyte origin [101,102]. Woods and Tilly [106] reported improving pregnancy outcomes in women with a previous history of assisted reproduction failure. They obtained autologous mitochondria from oogonial stem cells (OSCs) isolated from cortical biopsies. The injection was made via ICSI. The therapy was called Autologous Germline Mitochondrial Energy Transfer (AUGMENT) [106,107]. It aroused controversy from the very beginning [108,109,110,111]. Experts in the field were raising numerous questions about OSC (e.g., how many of them are recovered, how does their amplification affect them, are their mitochondria healthy) and AUGMENT protocol (e.g., how many mitochondria are being transferred; does transfer result in a significant boost to ATP). The therapy was considered to be based on the insecurities of the current knowledge about the character of the putative OSC and benefits claimed for mitochondrial transfer [111]. Moreover, there were also accusations of improper conduct of a clinical trial [108]. However, AUGMENT was commercially launched, raising a known question once again—does commercialisation overtake science in reproductive medicine? The introduction of the procedure turned out to give no benefit to the patients. That, in consequence, led the OvaScience—once worth over a billion dollars—to debt of millions and a reduction in workforce of around 50%. Finally, the company was sued by shareholders and closed its operations [112,113]. The AUGMENT history shows that, even though the field of reproductive medicine is highly privatised, it must have the same rules as the rest—first of all, be guided by evidence-based policy. Currently, there is one registered trial that aims to evaluate the effect of mitochondrial transfer from bone marrow mesenchymal stem cells on the quality of oocytes; however, the results are unknown [114].

## 6. Summary

Mitochondria play a crucial role in human fertility and early embryo development as it has a great contribution to several vital processes. Despite its great potential of clinical application, the field of mitochondrial research is still not well established. Further studies are needed to fully evaluate the clinical usefulness of assessing and improving mtDNA quantity and function.

## Figures and Tables

**Figure 1 cells-11-00797-f001:**
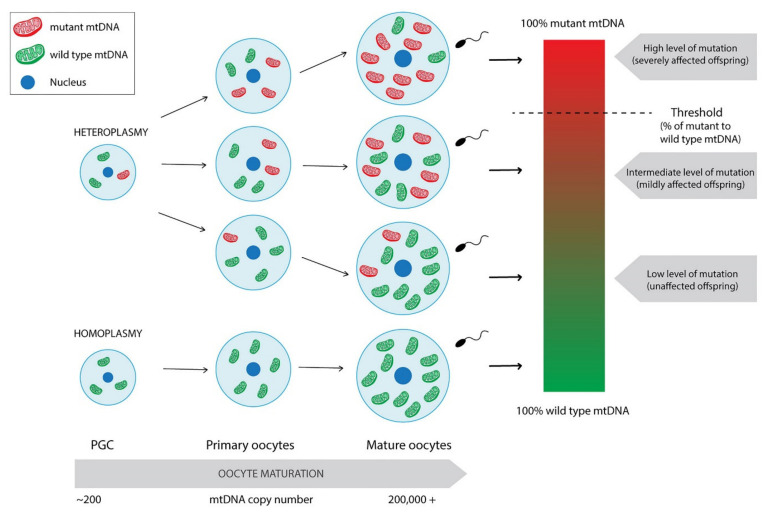
Differentiation of homoplasmic and heteroplasmic PGCs into mature oocytes with mtDNA copy number expansion. Mature oocytes derived from heteroplasmic PGC may present varying levels of mutation frequency leading to unaffected or mildly or severely affected offspring. Paternal leakage was not taken into consideration.

**Figure 2 cells-11-00797-f002:**
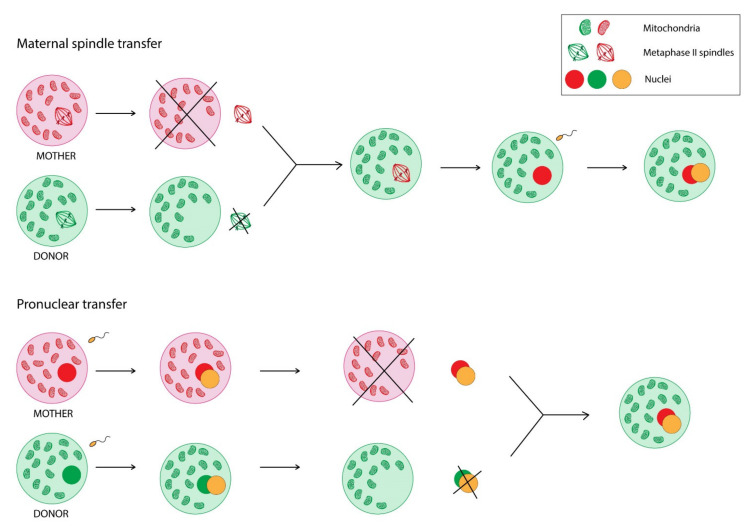
Comparison of two mitochondrial replacement technologies—maternal spindle transfer and pronuclear transfer.

**Table 1 cells-11-00797-t001:** Comparison of results obtained by several study groups evaluating the relationship of mtDNA content in trophoectoderm biopsy (TE) or blastomeres (B) with embryos’ features as aneuploidy, morphology, implantation, live birth and maternal age. Legend: N sample size, • positive correlation, • negative correlation, • no statistically significant correlation, - parameter not assessed.

	Material	N	Aneploidy	Morphology	Implantation	Live Birth	Maternal Age
Ritu et al., 2019	TE	287	•	•	•	•	•
Scott et al., 2020	TE	615	-	•	•	•	•
Wu et al., 2021	TE	1301	-	•	•	-	•
El-Damen et al., 2021	TE	355	-	•/•	•	•	•
Lee et al., 2019	B	39	•	-	-	-	-
	TE	998	•	-	•	-	•
De Munk et al., 2021	B	112	•	-	-	-	•
	TE	112	•	-	-	-	•
Diez-Juan et al., 2015	B	205	-	•	•	-	-
	TE	65	-	•	•	-	•
Arnanz et al., 2020	TE	504	•	•	-	-	-
Boynukalin et al., 2020	TE	707	-	-	-	•	-
Du et al., 2021	TE	246	•	•	•	-	-
Wang et al., 2021	TE	769	-	•	•	•	•
Klimczak et al., 2018	TE	1510	-	•	•	-	•
de Los Santos et al., 2018	TE	465	•	•	-	-	•
Victor et al., 2017	TE	1396	•	-	•	-	•
Fragouli et al., 2015	B	39	-	-	-	-	•
	TE	340	•	-	•	-	•
Fragouli et al., 2017	TE	199	-	-	-	-	•
Ravichandran et al., 2017	TE	1505	-	•	•	-	•
Treff et al., 2017	TE	374	-	•	•	-	•
Shang et al., 2018	B	149	•	•	•	-	•
	TE	250					
Podolak et al., 2022	B	314	•	•	•	•	•

## Data Availability

Not applicable.

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
