# Peer review of "The Role of Mitochondria in Human Fertility and Early Embryo Development: What Can We Learn for Clinical Application of Assessing and Improving Mitochondrial DNA?"

_cells, 2022, doi:10.3390/cells11050797_

Round 1

Reviewer 1 Report

The review by Podolak et al. represents a nice summary of literature on mtDNA content (especially in the context of quantity) and fertility.

The manuscript is mostly well written, and I recommend it for publication. There are only a few comments that need attention, as detailed below:

  • In the abstract the authors write ‘As mitochondrial DNA is exclusively maternally inherited, oocyte’s mitochondrial DNA level is crucial to provide sufficient ATP content for the developing embryo until the blastocyst stage of development.’. Later in the manuscript the authors report cases of paternal mtDNA transmission, though, this is not a general mechanism. The authors should therefore tone down their statement by e.g. adding the word ‘generally’.
  • In the introduction, line 35, the authors describe the length of mtDNA with 16.6 kb. However, this only refers to human or primate mtDNA, as mouse mtDNA has e.g. a length of 16.3 kb. Please clarify.
  • The paragraph starting with line 51 needs to be rewritten as it contains misleading information. The authors should consider obtaining the information from original literature instead of reviews reporting wrong statements themselves. E.g. in lines 58, 59, and 60 they write: ‘16,6 kbp MtDNA controls the synthesis of about 67 proteins, including 13 of the OXPHOS, while the rest of the bacterial genes were transferred to the nuclear genome [3,7,8].’. This is not true and was already misinterpreted in the cited review paper. The mentioned 67 proteins are encoded by the mitochondrion with the largest coding repertoire, that of Reclinomonas americana. Human mtDNA (which has a length of ~16.6 kb) only encodes 13 proteins.
  • In lines 142, 143, and 144 the authors state ‘Reaching the threshold of mutation frequency leads to mitochondrial dysfunction and thus, severely affected offspring (Figure 1) [30,31].’. Not every mutation/variant in mtDNA leads to mitochondrial dysfunction. Among other, there are e.g. variants defining the different mtDNA haplogroups which are not associated with any disease or synonymous mutations. To avoid misunderstanding, such information should be added to the text.
  • Regarding the bottleneck theory (paragraph starting with line 159): The reduction in mtDNA number in PGCs is not the only factor that can contribute to the observed germline mtDNA bottleneck, as e.g. discussed in a review by Stewart and Chinnery (Nature, 2015, https://www.nature.com/articles/nrg3966.pdf). According to shifts in frequency of heteroplasmies from one generation to the next, the size is actually much smaller that the ~200 mtDNA copies in PGCs. To avoid wrong conclusion, the authors should include such information.
  • In lines 249 and 250 the authors write: ‘As the mitochondrial genome has only exons and no introns, any point mutation or deletion could disrupt cellular respiration [58].’ This sentence is misleading as someone could think that mtDNA only consists of exons, while there is also non-coding DNA (D-loop). Please clarify.
  • In the legend of Table 1, please explain what N refers to.
  • There are still several formatting issues in the text such as double-spaces or reduced font size of some paragraphs.

Reviewer 2 Report

The review article “The role of mitochondria in human fertility and early embryo development: what can we learn for clinical application of assessing and improving mitochondrial DNA?” is dedicated to the analysis of the role of mitochondria and mitochondrial DNA in human reproduction and the utility of their clinical use.

The article is well written.

The study has a good design.

The article is logically divided into sections and subsections.

In the article there are no grammatical and stylistic errors.

There is a table and many figures of good quality presented in the article.

The references cited are relevant and adequate.

A large number of scientific literature sources were analyzed.

The work has a high degree of novelty.

In my opinion, this review paper can be recommended for publication after minor revision.

It is recommended to expand section “Impact of mitochondrial insufficiency on fertility.”

It is recommended to include a list of abbreviations, used in the article.
